# Selection of New Heat Treatment Conditions for Novel Electroless Nickel-Boron Deposits and Characterization of Heat-Treated Coatings

**Véronique Vitry** [1,*] **, Muslum Yunacti** [1,*] **, Alexandre Mégret** [1] **, Hafiza Ayesha Khalid** [1,2] **, Mariana Henriette Staia** [3,4] **and Alex Montagne** [5]

1   Metallurgy Department, University of Mons, 20 Place du Parc, 7000 Mons, Belgium
2   Materials Science Department, Faculty of Engineering, University of Mons, 20 Place du Parc, 7000 Mons, Belgium
3   School of Metallurgical Engineering and Material Science, Faculty of Engineering, Universidad Central de Venezuela, Los Chaguaramos, Caracas 1040, Venezuela
4   Venezuelan National Academy for Engineering and Habitat, Palacio de las Academias, Caracas 1010, Venezuela
5   Arts et Métiers Institute of Technology, MSMP, HESAM Université, F-59000 Lille, France
*   Correspondence: veronique.vitry@umons.ac.be (V.V.); muslum.yunacti@umons.ac.be (M.Y.); Tel.: +32-4-6705-7882 (M.Y.)

**Abstract:** Electroless nickel-boron coatings were deposited from a novel deposition bath that was stabilizer-free. Those coatings were analyzed by DSC to determine the crystallization temperature of nickel-boride phases, and then the best heat treatment conditions for the new coatings were selected using the Knoop hardness test and XRD. The results of DSC analysis and XRD were coherent with the findings of previous studies, which shows that the new coating has a boron content lower than 4% wt. The maximum hardness was obtained after heat treatment at 300 °C for 4 h and reached $1196 \pm 120$ hk$_{50}$, which is much higher than the as-deposited coating. The heat-treated coatings were then fully characterized using optical and scanning electron microscopy, as well tribological and corrosion tests. Various sliding tests (ball-on-disk and ball-on-flat configurations) were conducted to analyze the coefficient of friction (COF) and wear behavior of the coatings. The maximum von Mises stress was calculated, and values of 624 MPa and 728 MPa were obtained for the ball-on-disk and ball-on flat, respectively, at a depth close to 14 μm from the surface, confirming the suitability of the applied load. Abrasive, adhesive, and fatigue wear mechanisms were observed on the worn sample morphology using SEM. It has been determined that during the corrosion test the OCP and corrosion potential values for the heat-treated coating increase as compared with the as-deposited one, whereas its corrosion resistance decreases.

**Keywords:** electroless nickel-boron deposits; structure; DSC; XRD; heat treatment; hardness

## 1. Introduction

Electroless nickel coatings have been the object of numerous studies and developments due to their interesting features such as easy plating process, low cost, high deposition rate, uniformity of the formed coatings, excellent hardness, and good corrosion and wear resistance [1–7]. The method used to form them, electroless plating, is based on the autocatalytic reduction of metallic ions in an aqueous solution by a chemical agent present in the solution (the reducing agent) [4,8,9]. The electroless bath contains, next to the metallic ions and reducing agent, a complexing agent aimed at maintaining the concentration of the free metallic ion constant, a stabilizer, and a buffering agent. The nature and concentration of those agents have a significant impact on the composition and properties of the formed coatings [1,9].

Among electroless nickel coatings, borohydride-reduced coatings present specifically attractive properties, leading to sustained attention both in research and industry [10–14]. Their main advantages are their superior hardness and excellent wear resistance [1,11,15–21], which exceed those of hard chrome and tool steels [1,5,11]. Electroless nickel-boron coatings are used to increase the wear and corrosion resistance of various pieces of equipment, such as tools, valves, pumps, gear, and pipelines, which are used in several industries, such as automotive and oil [1,14,22,23]. Moreover, the inherent self-lubricious morphology of the electroless nickel-boron (ENB) coatings makes them appropriate for applications where components are subjected to wear [24].

Like all electroless nickel coatings, the mechanical and tribological properties of electroless nickel boron can be enhanced by heat treatments [15,25–27] due to the formation of crystalline nickel and nickel-boride phases [13], and hardness close to 1200–1300 $VHN_{100}$ is observed after adequate heat treatment [28,29]. The phases that are present in the coating after heat treatment and the temperature at which those are formed vary significantly with the composition of the coating: coatings with a lower boron content (up to 5.8 wt. %) present crystallization of $Ni_3B$ around 300 °C, while coatings with a higher boron content also present crystallization of $Ni_2B$ (around 400 °C) [30]. Those crystallization phenomena affect the mechanical, tribological, and corrosion resistance properties of electroless nickel boron [7].

The properties of heat-treated coatings are not only affected by composition: heat treatment temperature and dwelling time are also significant parameters [19,31]. It is widely accepted that the hardness of heat-treated coatings increases with heat-treated temperature up to approximately 400 °C, then decreases with further temperature increase [13,30,32]. Likewise, the hardness of coatings increases initially with dwelling time, but decreases after a threshold is reached (that varies with treatment temperature) [32].

Standard electroless nickel-boron coatings, despite their excellent features, suffer from one main disadvantage, which is the presence of lead (used for stabilization) in the plating bath, which is then incorporated into the coatings. The metallurgy lab of UMONS has worked to develop coatings exempt of this heavy metal [33,34] and has developed a novel electroless plating bath that does not include a stabilizer by modifying the concentration of complexing and reducing agents [35]. In a previous study [35], the novel coating was comprehensively compared with the traditional electroless nickel-boron coating. The novel coating does present modified properties as compared to the electroless nickel-boron coatings: its surface is smooth and featureless, its boron content is lower, and its corrosion resistance is improved while maintaining the same hardness. The enhancement in the corrosion resistance is related to the morphology of the coating: the new coating presents a column-free morphology that does not provide a path for the corrosive solution to reach the substrate, while the standard electroless nickel coating has a columnar morphology [36].

Due to those differences, and mainly the lower boron content, it is necessary to investigate the behavior of those coatings during heat treatments, on one hand, to determine the heat treatment conditions that lead to the optimal hardness, and on the other hand to determine if the coatings treated in the determined conditions reach industrial requirements in terms of hardness and wear and corrosion resistance.

This study is divided into two parts: First, the heat treatment behavior of the coatings is assessed, and the hardness of coatings treated in various conditions of temperature and dwelling time was determined in order to determine the relation between annealing time and temperature and hardness. Second, the corrosion and tribological performance of the heat-treated coating that presents the highest hardness were investigated using potentiodynamic polarization, neutral salt spray tests, and continuous sliding (ball-on-disk) and reciprocal (ball-on-flat) wear tests. The latter was used to enable future characterization of the coatings by tribocorrosion tests.

## 2. Materials and Methods

### 2.1. Substrate Preparation

In this study, ST 37-DIN 17100 mild steel with dimensions of $50 \times 25 \times 1$ mm$^3$ was used as a substrate. SiC paper with 180, 500, and 1200 grit was used for substrate grinding. The substrates were then cleaned in acetone. The last preparation step was their activation in 30 vol. % HCl for 3 min. After activation, the substrates were immersed directly in the deposition bath for one hour. Substrates were rinsed with distilled water after each step.

### 2.2. Electroless Nickel-Boron Plating and Heat Treatment

The electroless nickel-boron deposition bath is shown in Table 1. The deposition was carried out with a volume of 1 L, on a regulated hot plate with magnetic stirring. The process temperature was fixed at $95 \pm 1$ °C. After the 1 h-process, a nickel-boron deposit with 15 μm thickness was obtained [35].

**Table 1.** Composition of electroless nickel-boron deposition bath.

| Compound | Amount [35] |
|---|---|
| NiCl$_2$·6H$_2$O (99%—VWR Chemicals) (g/L) | 24 |
| NaBH$_4$ (99.9%—Acros Organics) (g/L) | 0.4 |
| NH$_2$-CH$_2$-CH$_2$-NH$_2$ (99% VWR Chemicals) (mL/L) | 120 |
| NaOH (VWR Chemicals) (g/L) | 160 |

The coated samples were heat-treated in a 95% Ar—5%H$_2$ atmosphere at a pressure of 1 bar in a TermConcept tubular furnace. To investigate the effect of the heat treatment parameters, samples were heat-treated with various temperature/time couples, as shown in Table 2.

**Table 2.** Heat treatment temperature/time couples applied to the heat treatment of electroless nickel-boron coatings.

| Temperature (°C) | Time (h) | | | |
|---|---|---|---|---|
| | 0.5 | 1 | 2 | 4 |
| 250 | - | ✓ | - | - |
| 300 | - | ✓ | ✓ | ✓ |
| 350 | - | ✓ | ✓ | ✓ |
| 400 | - | ✓ | ✓ | - |
| 450 | ✓ | ✓ | - | - |

### 2.3. Characterization

DSC with a heating rate of 10 °C/min in N$_2$ atmosphere was used in the continuous heating mode up to 600 °C to determine the crystallization onset temperature of the coating.

A Hitachi SU8020 (Tokyo, Japan) SEM was used to analyze the top surface and cross-section of the samples. Cross-sections of the samples were prepared by mounting cut samples in a cold resin and grinding with SiC paper (from 180 to 4000, 3 min each). Subsequently, the samples were polished, firstly with diamond paste (1 and 3 μm, 5 min each) up to mirror finish, then with a colloidal silica suspension for 3 min. Finally, the specimen sections were etched with Nital 10% for 2 min.

XRD analysis was carried out with a Siemens D50 spectrometer in θ–θ configuration. The measurements were carried out with copper Kα radiation (λ Kα = 1.54 Å). The scan angle ranged from 30° to 70° with a scan rate of 0.017°/s.

A Mitutoyo HM-200 (Kawasaki, Japan) microhardness tester was used to determine the hardness of the deposits. The hardness of the samples was measured by Vickers microindentation on the top surface, with a load of 0.5 N and a dwelling time of 20 s.

Knoop microindentation was carried out on the cross-section using a load of 0.2 N and 0.5 N and a dwelling time of 20 s. The reported values are the average of ten measurements per sample.

Young's modulus and hardness of the samples were measured by instrumented indentation testing (IIT) using an XP nanoindenter (MTS, Eden Praire, MN, USA) equipped with a Berkovich indenter running in continuous stiffness measurements (CSM) mode with an imposed maximum penetration depth of 100 nm reached and a constant strain rate of $0.05 \text{ s}^{-1}$. The tip area function was calibrated using fused silica. The Oliver and Pharr method [37] was chosen for the analysis of the load-displacement curves. Nanoindentation tests were carried out on mounted and mirror-polished cross-sections that were prepared as described above. An array of 100 indents was performed on the cross-section of the mounted sample covering different parts (resin, coating, and substrate). This was repeated for each sample.

A Zeiss 119 Surfcom 1400D-3DF (Oberkochen, Germany) apparatus and Zeiss brand software were employed to analyze the deposit surface roughness. Three different roughness values, namely average (Ra), peak (Rp), and valley (Rv) roughness, were investigated. Ten measurements were conducted on the samples, and the average of them was reported.

A ball-on-disk Tribotechnic microtribometer and a ball-on-flat Bruker tribometer were used to investigate the tribological behavior of the deposits. The tests were employed without lubrication. Alumina balls were used as counterparts for both tests. During the tests, COF was simultaneously recorded. The conditions for both ball-on-disk and ball-on-flat reciprocal sliding wear tests are presented in Table 3. Both a Hitachi SU8020 SEM and a HIROX KH-870 digital optical microscope were used for the surface analyses after each test. Energy-dispersive X-ray spectroscopy (EDS)–Hitachi SU8020 SEM was employed to determine the surface composition of the wear tracks and balls after the ball-on-flat and ball-on-disc wear tests.

**Table 3.** Parameters of sliding wear tests.

| Parameters / Test | Ball-on-Disk | Ball-on-Flat |
|---|---|---|
| Load (N) | 2 | 2 |
| Sliding speed/frequence | 10 cm/s | 5 Hz |
| Sliding distance (m) | 1000 | 1000 |
| Counter body | $Al_2O_3$ | $Al_2O_3$ |
| Counterbody diameter (mm) | 6 | 4.8 |
| Wear track (mm) | 3 (radius) | 10 (length) |
| Environment | Ambient | Ambient |

Potentiodynamic polarization curves were obtained in a 0.1 M NaCl solution using a Bio-Logic (Claix, France) SP-300 potentiostat. A standard three-electrode cell was used to perform the tests. The sample, a platinum plate and Ag/AgCl (KCl-saturated) were used as the working electrode, counter electrode, and reference electrode, respectively. The potential range and scan rates were $\pm 0.25$ V vs. OCP and 1 mV/s, respectively.

A Q-FOG cyclic corrosion tester (Q-Lab, Westlake, OH, USA) was used for neutral salt spray tests according to the ASTM B117-07 standard. The surface exposed to salt spray was the same for all samples. The surface exposed to the saltwater was a circle with a 0.6 cm radius. The samples were suspended in the cabinet for 10 days with a period of 1 h, 4 h, 8 h, 1, 2, 3, 4, and 5 days of control and image taking. To quantify the corroded area after the salt spray tests, ImageJ, an open-source image processing analysis program, was used.

## 3. Results and Discussion

### 3.1. Determination of the Crystallization Onset Temperature

DSC studies were carried out to assess the crystallization temperature of the nickel-boron coatings. The DSC trace is exhibited in Figure 1, and it shows only one exothermic peak at 284 °C that is attributed to the precipitation of the orthorhombic $Ni_3B$ phase,

which is expected for a coating with a boron content less than 5.8 wt. %, whereas deposits containing more than 5.8 wt. % B are expected to exhibit two exothermic peaks: one at around 300 °C (linked to $Ni_3B$ formation) and the other at around 430 °C (linked to $Ni_2B$) [2,32,38,39]. In the present study, the deposit analyzed by DSC has 4 wt. % B [35], which explains the presence of only one peak on the DSC trace. It can be concluded that the results of DSC analyses confirm the GDOES findings reported in the previous study [35]. Based on those results, it is possible to confirm the temperature range for heat treatment (from 250 to 450 °C).

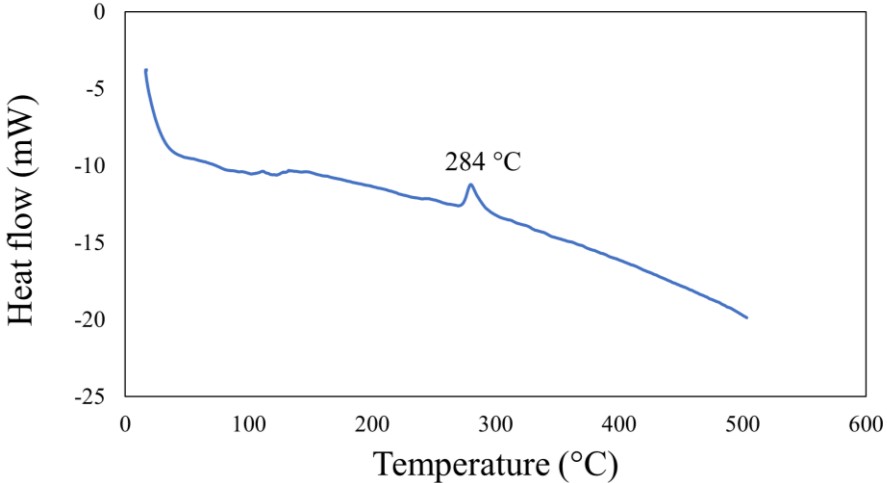

**Figure 1.** DSC trace of the ENB deposit.

### 3.2. Structural Analysis

The XRD results of the heat-treated coatings are shown in Figure 2. The XRD pattern of the as-deposited sample exhibits a broad hump characteristic of amorphous phases as reported by other researchers [16,40–42]. A small peak at 65° corresponding to the ferrite phase of the steel substrate is also observed, which is normal because the coating thickness is not enough to hinder the influence of the substrate. Heat treatment higher than 250 °C for 1 h shows the appearance of crystalline $Ni_3B$ with a weak Ni peak at 51.8°. The formation of crystalline $Ni_3B$ is in agreement with the Ni-B phase diagram [43] and the finding from the DSC analysis presented above. In addition, $Ni_2B$ phases are also present in the XRD spectra of the samples heat-treated at 400 °C and 450 °C.

### 3.3. Effect of Heat Treatment Temperature and Time on Hardness

Figure 3 illustrates the hardness evolution after heat treatment conducted at different annealing temperatures for different times. After heat treatment at 250 °C and above, the deposits exhibit higher hardness as compared to the as-deposited one, which is a consequence of the formation of the $Ni_3B$ phase, which presents a high hardness [30,40,44]. The maximum hardness of 1300 $hk_{20}$ is obtained for samples heat-treated between 300 °C and 400 °C, which agrees with the findings reported by several researchers for other electroless nickel-boron compositions [13,15]. The deposit annealed at 450 °C has a lower hardness compared to the one annealed at 400 °C. This has also been reported by several researchers [32,41,45,46] and is attributed to an increase in grain size.

The deposits were also annealed at 300 °C, 350 °C, and 400 °C for varying times (0.5, 2, and 4 h). The results are shown in Figure 3. The hardness of the deposit annealed at 300 °C increased after increasing the annealing time from 1 h to 4 h. This rise could be due to the complete crystallization of $Ni_3B$. The sample annealed at 300 °C for 4h exhibits the highest hardness among all the samples. Similar results were reported by Pal et al. [32], who found an increase in hardness up to 15 GPa after annealing at 300 °C for 5 h for lead-stabilized electroless nickel boron. However, in the case of annealing at 350 °C and 400 °C, the hardness of the deposits decreased with an increase in the annealing time, which can be

explained by the formation of soft crystalline Ni in the deposit [32] and an increase in the grain size. As the maximum hardness was obtained for the samples heat-treated at 300 °C for 4 h, further characterization and comparison with the as-deposited condition were carried out exclusively.

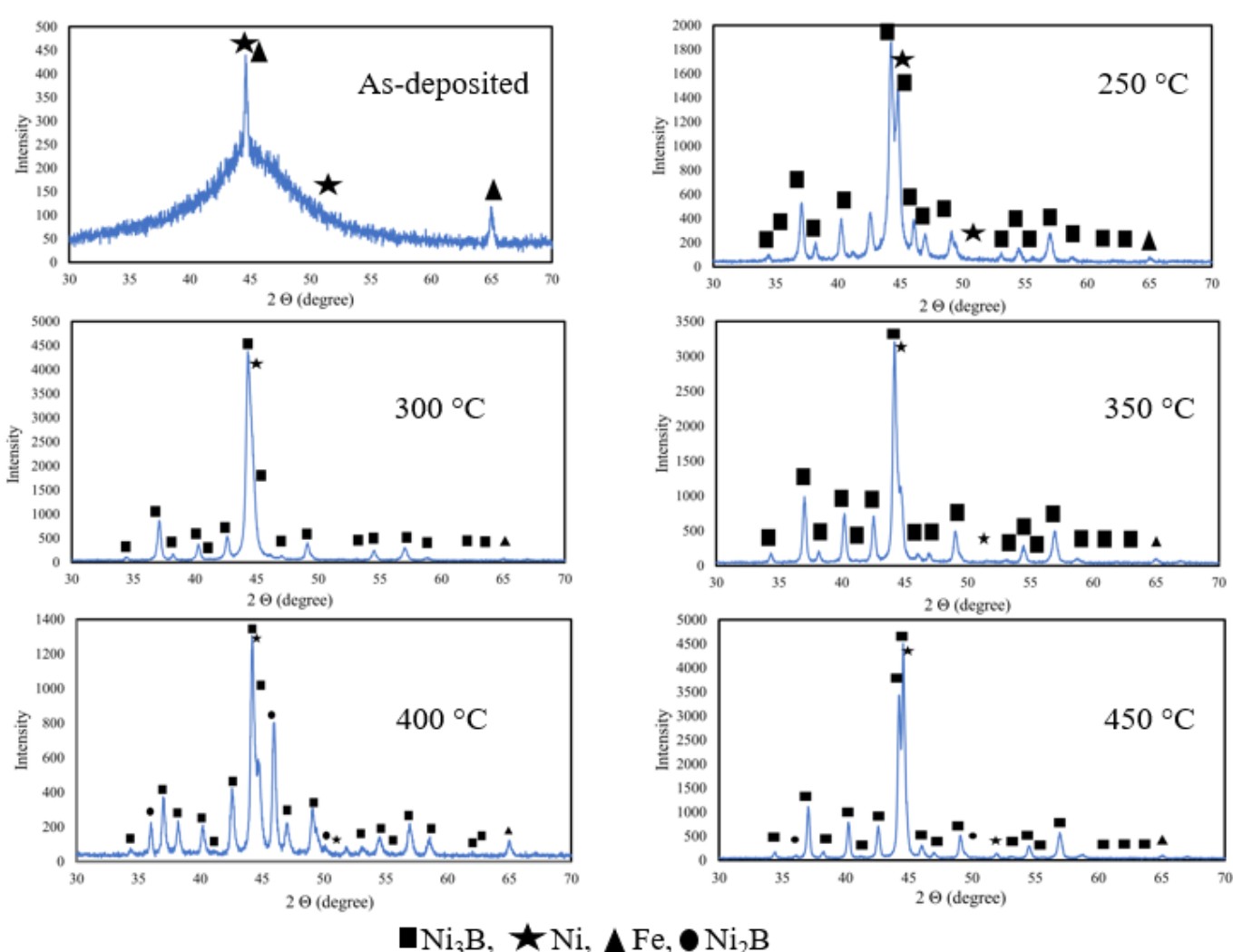

**Figure 2.** XRD data of the ENB deposits before and after heat treatment at 250 °C, 300 °C, 350 °C, 400 °C, and 450 °C for 1 h.

### 3.4. Surface and Cross-Section Morphology of the Coatings with Optimized Heat Treatment

The cross-section and surface morphology of the deposit heat-treated under optimum conditions (300 °C, 4 h) is shown in Figure 4. The cross-section is featureless and the surface morphology exhibits a smooth surface appearance, similar to the as-deposited samples [35], in agreement with the results reported by several researchers [15,16,41]. Heavy metal stabilizers deposit on the substrate and co-deposit with nickel, causing columnar morphology, and their absence produces a smooth surface [7].

### 3.5. Mechanical Properties

Knoop cross-section microhardness has been used for the screening of the best heat-treatment conditions, but it was supplemented with other techniques (IIT and Vickers surface microindentation) for this full characterization. Moreover, IIT also allowed the determination of Young's modulus of the deposits. In Table 4, the results were compared with those obtained for the as-deposited samples. The hardness of the coatings increased by around 40 % after heat treatment. As aforementioned, this increase is due to the

precipitation of $Ni_3B$ [47,48]. Young's modulus is also increased after heat treatment, which is in agreement with the findings obtained by several researchers for other electroless nickel-boron compositions [28,47,49].

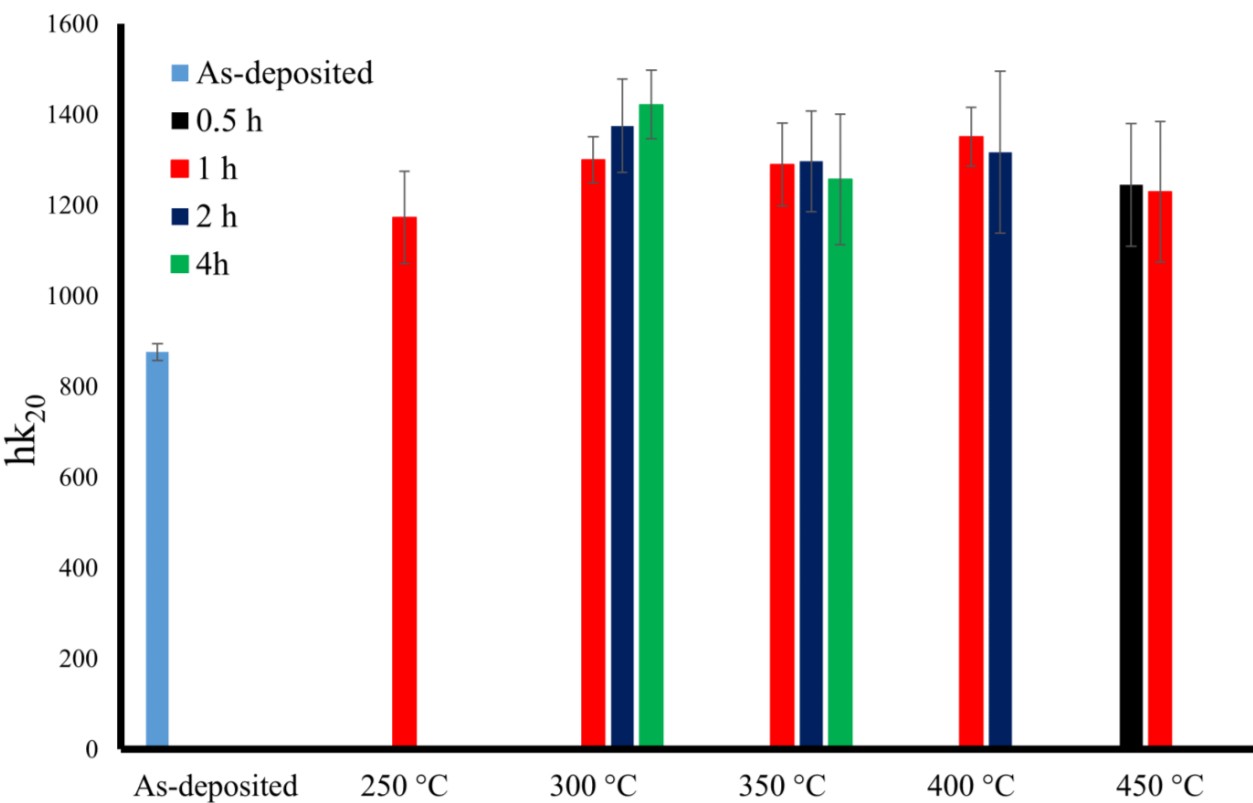

**Figure 3.** Cross-section hardness of the ENB coating in the as-deposited condition and after heat treatment at 250 °C, 300 °C, 350 °C, 400 °C, and 450 °C for 1 h, 2 h, 4 h, and 0.5 h.

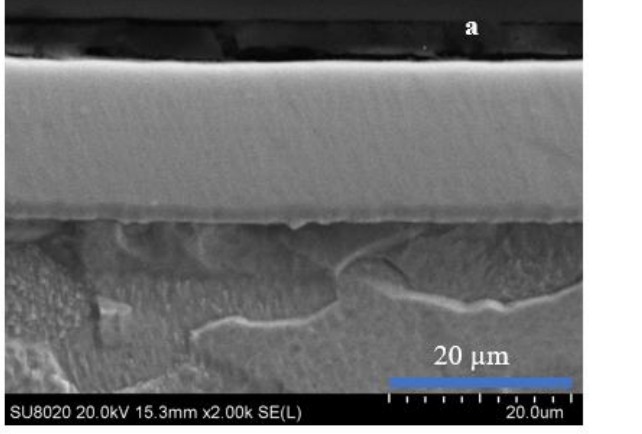
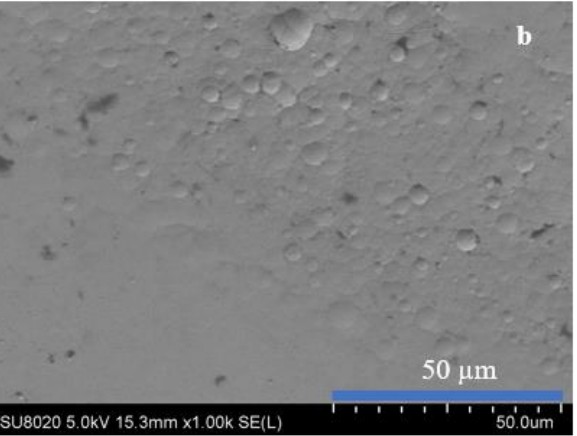

**Figure 4.** Cross-section (**a**) and surface (**b**) morphologies of the ENB deposit annealed at 300 °C for 4 h.

Figure 5 presents the evolution of the hardness and Young's modulus of the heat-treated deposit, as a function of the distance from the substrate/coating interface. The continuous red lines and the dashed lines represent the average value and standard deviation, respectively. First of all, for both Young's modulus and hardness, the deposits present homogeneity throughout the deposit thickness.

**Table 4.** Comparison of the mechanical properties of samples in the as-deposited and optimized heat-treated conditions.

| Hardness | Heat-Treated Sample | As-Deposited Sample [30] |
|---|---|---|
| Knoop hardness ($hk_{50}$) | $1196 \pm 120$ | $886 \pm 30$ |
| Vickers hardness ($hv_{50}$) | $1277 \pm 181$ | $933 \pm 62$ |
| IIT Hardness (GPa) | $16.2 \pm 3.0$ | $11.6 \pm 0.3$ |
| Young's modulus (GPa) | $277 \pm 26$ | $201 \pm 10$ |

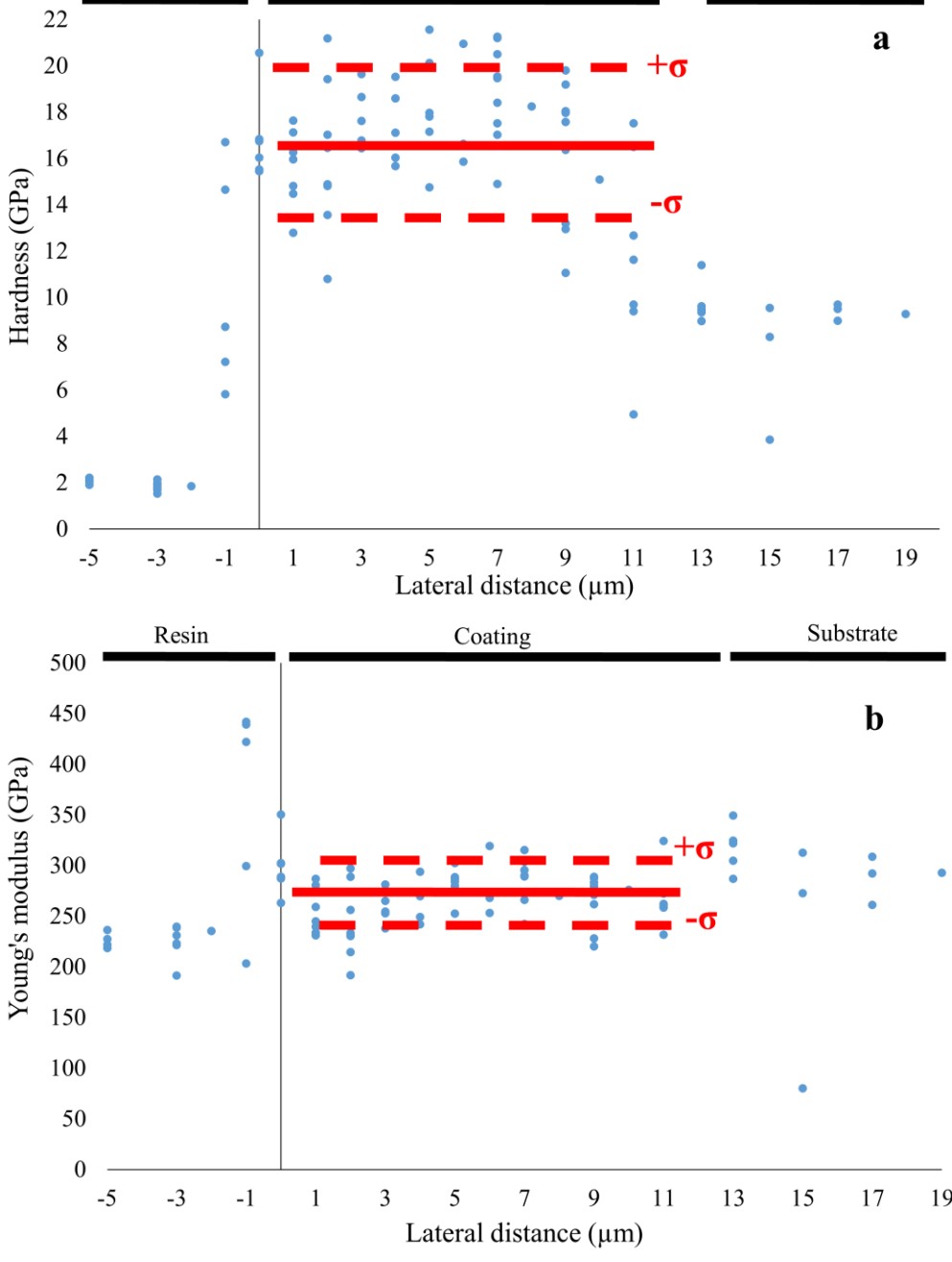

**Figure 5.** Hardness (**a**) and Young's modulus (**b**) of the heat-treated deposit. The average value is drawn (continuous red line), as well as the standard deviation (dashed red line).

The standard deviation in the hardness after heat treatment is higher than the one of the as-deposited [35]. This could be due to the formation of different crystalline nickel

phases and nickel-boron phases and their distribution in the coating. The second reason is related to the size of the imprint, which is decreased with the increase in hardness, thus naturally increasing the standard deviation.

### 3.6. Roughness

The roughness values of the deposits before and after heat treatment are presented in Figure 6. It is observed that the roughness values have not significantly changed after heat treatment, which is expected because the surface morphology (that is strongly interlinked with roughness) is kept unmodified after heat treatment. It is also found that Rp (peak roughness) is found to be higher than Rv (valley roughness). This could be due to the absence of columns: there is no hole in the coating, only protrusions (see Figure 6).

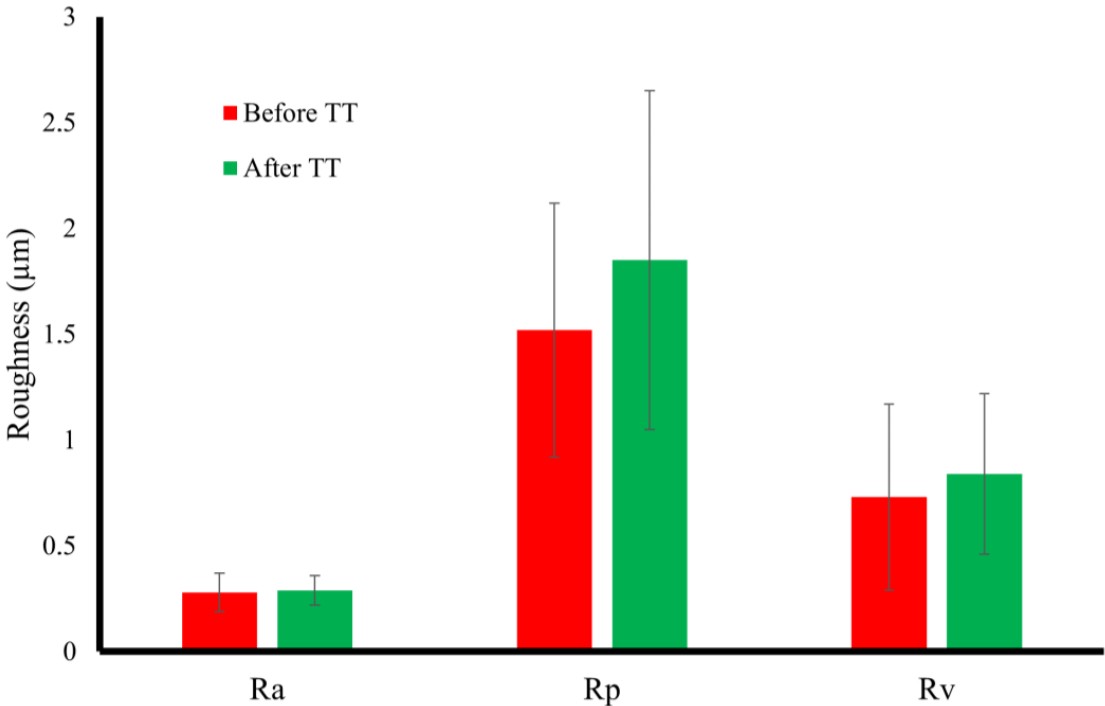

**Figure 6.** The average (Ra), peak height (Rp), and valley depth (Rv) roughness of the deposits before and after heat treatment.

### 3.7. Tribological Properties

#### 3.7.1. Stress Analysis Prior to the Wear Test

In the previous publication [35], the authors indicated that by applying Hertz formulation it is possible to predict the maximum von Mises stress and its location in an ENB as-deposited coating as a function of the applied load. It was shown that for a coating of 15 μm thickness, the maximum von Mises stress will be located within the coating. Therefore, the substrate will not undergo plastic deformation and the coated system will maintain its integrity. In the present research, the elastic properties of the heat-treated coating are shown in Table 5. By applying a load of 2 N on the alumina balls of two different diameters, depending on the wear test (6 mm for ball and disc and 4.8 mm for the flat-on-disc tests, respectively), maximum contact pressures of approximately 1.07 GPa and 1.2 GPa, respectively, are calculated. As shown in Figure 7, for each test configuration the maximum von Mises stress, $\sigma_{max}$, achieves a value of 624 MPa and 728 MPa, respectively. These stresses are predicted to occur at a depth somewhat less than 14 μm from the coating surface, i.e., inside the coating, whose yield stress of 5400 MPa is much higher than the calculated $\sigma_{max}$.

**Table 5.** Data needed to calculate $P_{max}$ using Hertz equations.

| Properties<br>Material | E (GPa) | Thickness (µm) | Poisson's Ratio (ν) | Yield Stress<br>σ (MPa) |
|---|---|---|---|---|
| Heat-treated deposit | 277 | 15 | 0.31 | 5400 |
| Steel | 210 | | 0.33 | 400 |
| Al$_2$O$_3$ | 360 | | 0.2 | |

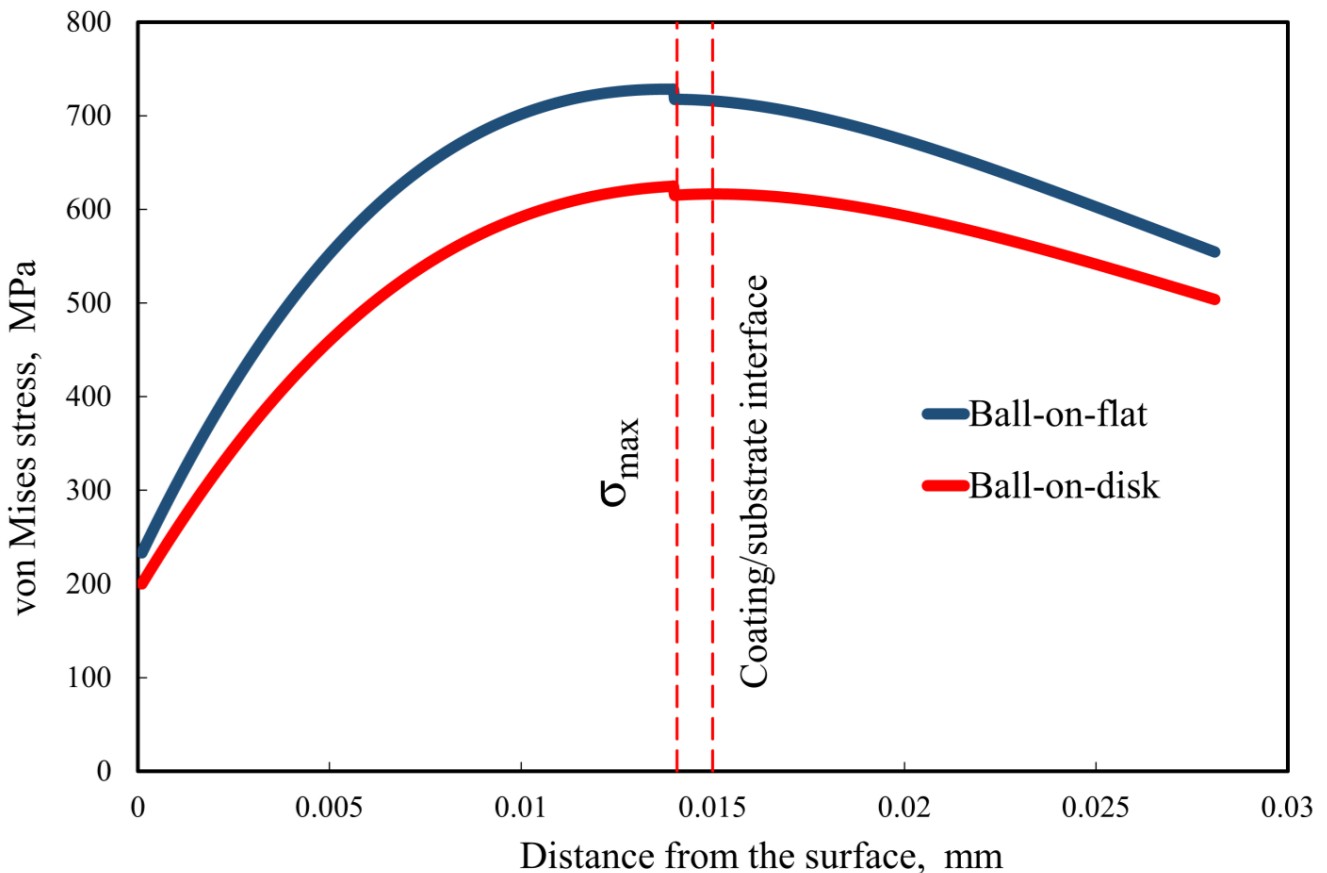

**Figure 7.** Change in the von Mises Stress for the heat-treated coating during both the ball-on-disk and ball-on-flat sliding tests.

### 3.7.2. Ball-on-Disk Sliding Wear Test

The COF recorded during the sliding tests is presented in Figure 8. It is clear that the COF of the heat-treated deposit is slightly higher than the one corresponding to the as-deposited coating in the same conditions of testing. This increase in the COF value is attributed to the presence of harder and small inter-metallic boride particles precipitated during the crystallization of the amorphous coatings in the Ni matrix, as a consequence of the heat treatment process. Their presence in the tribocontact between the hard ball and the coating contributes to an increase in its wear rate due to both a three-body wear abrasion and adhesive mechanisms.

The wear track of the heat-treated coating after testing is shown in Figure 9a,b. Longitudinal micro-grooves can be seen along the sliding direction. Well-defined grooves with ridges are observed in Figure 9a,b, as well as some patches of adhered material.

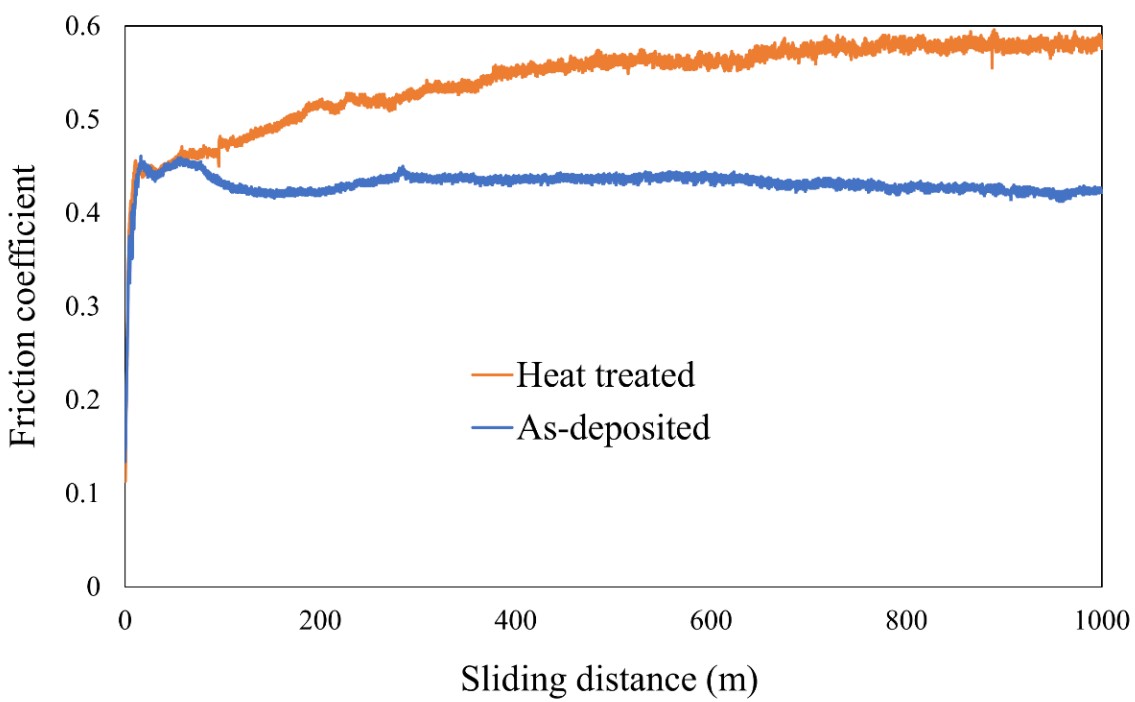

**Figure 8.** Evolution of friction coefficient of the as-deposited and heat-treated samples determined during a 1000 m of ball-on-disk sliding test.

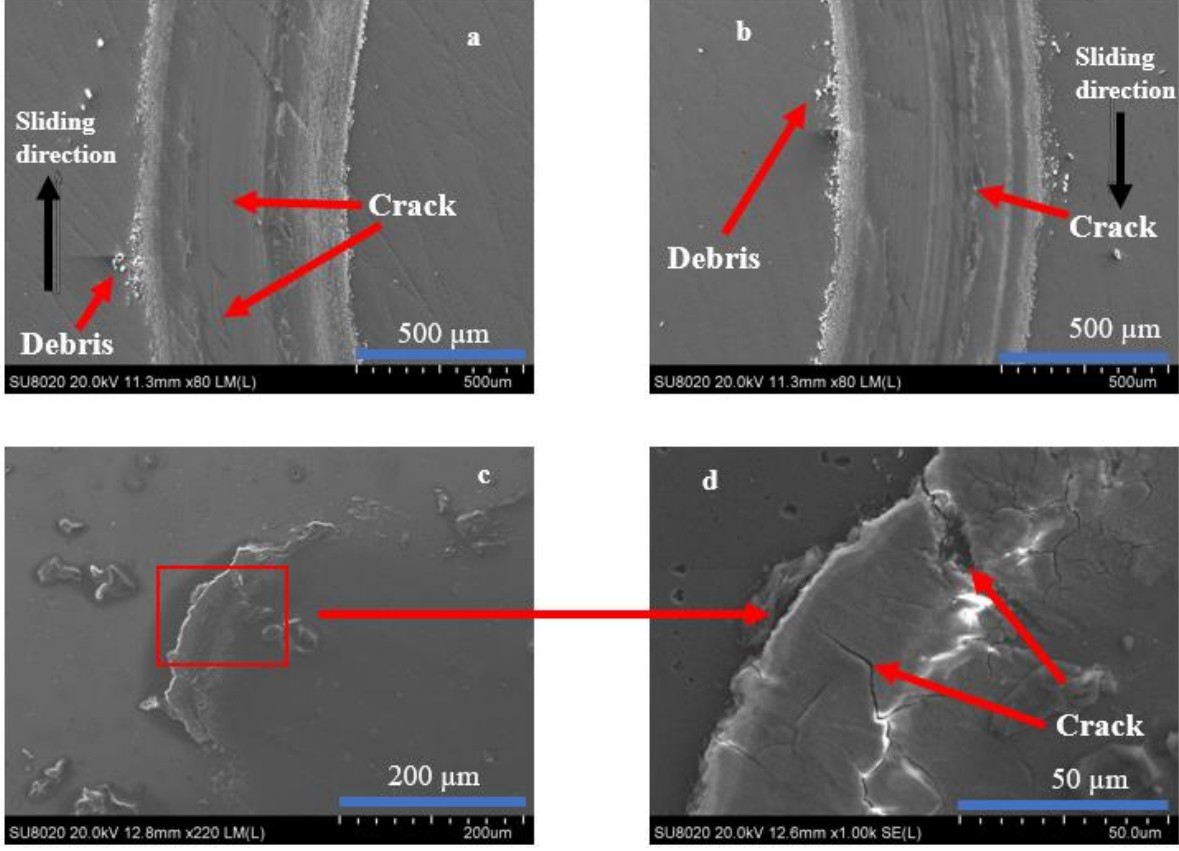

**Figure 9.** (**a**,**b**), SEM micrographs of the wear tracks from the heat-treated electroless nickel boron; (**c**,**d**), surface of the alumina balls after the dry ball-on-disk sliding test.

After analyzing the wear track of both the coating and ball, three different mechanisms were identified: abrasive, adhesive, and fatigue wear. The indications of abrasive wear are the formation of debris and signs of material loss on the wear track, especially on the middle of that corresponding to the coating. As shown in Table 6, the existence of nickel on the ball surface is due to the adhesion of the softer coating material onto the hard alumina ball. In Figure 9, some cracks are also indicated, which are formed as a consequence of material fatigue due to the cycling loading induced by the ball during the sliding test.

**Table 6.** Chemical composition of the wear track on the heat-treated deposit, debris, and ball after the dry ball-on-disk sliding test.

| Element | Oxygen (wt. %) | Aluminum (wt. %) | Iron (wt. %) | Nickel (wt. %) |
|---|---|---|---|---|
| Wear track on deposit | 5 | 0.4 | 0.6 | 93.8 |
| Debris | 5.4 | 0.5 | 0 | 93.6 |
| Wear track on the ball | 61.1 | 13.5 | 0 | 24.9 |

Chemical surface analysis carried out in the wear track, on the debris, and on the ball are presented in Table 6. It shows that no delamination of the coating took place during the test. The presence of oxygen in the wear track is considered to be due to surface oxidation caused by heating during the wear test. On the other hand, the existence of small amounts of iron in the wear track could be associated with the relatively low coating thickness.

3.7.3. Ball-on-Flat Sliding Wear Test

The evolution of the COF obtained during the ball-on-flat sliding test is presented in Figure 10. It is clear that both the heat-treated and the as-deposited samples present a similar friction behavior. Both COFs stabilize at approximately 0.58. The fluctuation in the COF of the as-deposited coating is slightly higher than that of the heat-treated coating, as a consequence of an increased amount of the adhesive mechanism present in the former one.

The wear track of the ENB-HT deposit after ball-on-flat is shown in Figure 11a,b. The wear track width exhibits a significant variation from one region to another. Delamination has not been observed after the test. However, as seen from the SEM micrograph of the track, slightly more debris were formed during the ball-on-flat test as compared to the ball-on-disk test. These results could be explained by taking into consideration that, for the former test, the half stroke length was higher by nearly 70% than the equivalent disc track radius. In addition, the ball diameter was smaller by 25% and the wear track length was nearly double. Under these conditions, it is expected that the sample tested in the ball-on-flat configuration is subjected to a higher mechanical solicitation than the sample tested in the ball-on-disk configuration.

The ball morphology was also investigated by means of SEM, and the corresponding micrographs are presented in Figure 11c,d. As in the previous configuration, a certain amount of nickel is present on the ball, which could be due to the adhesion of debris to the indenter.

The results of the surface chemical analysis carried out on the wear track, debris, and the ball are presented in Table 7. The results of this wear track are similar to those obtained in the wear track produced during the ball-on-disk test. Mostly nickel and a small amount of oxygen were determined. Iron was detected on the wear track of the deposit but not on the ball, as a consequence of the low coating thickness. Therefore, the detected iron belongs to the substrate and the fact that it was not found on the ball indicates that the coating maintained its mechanical integrity during the sliding test.

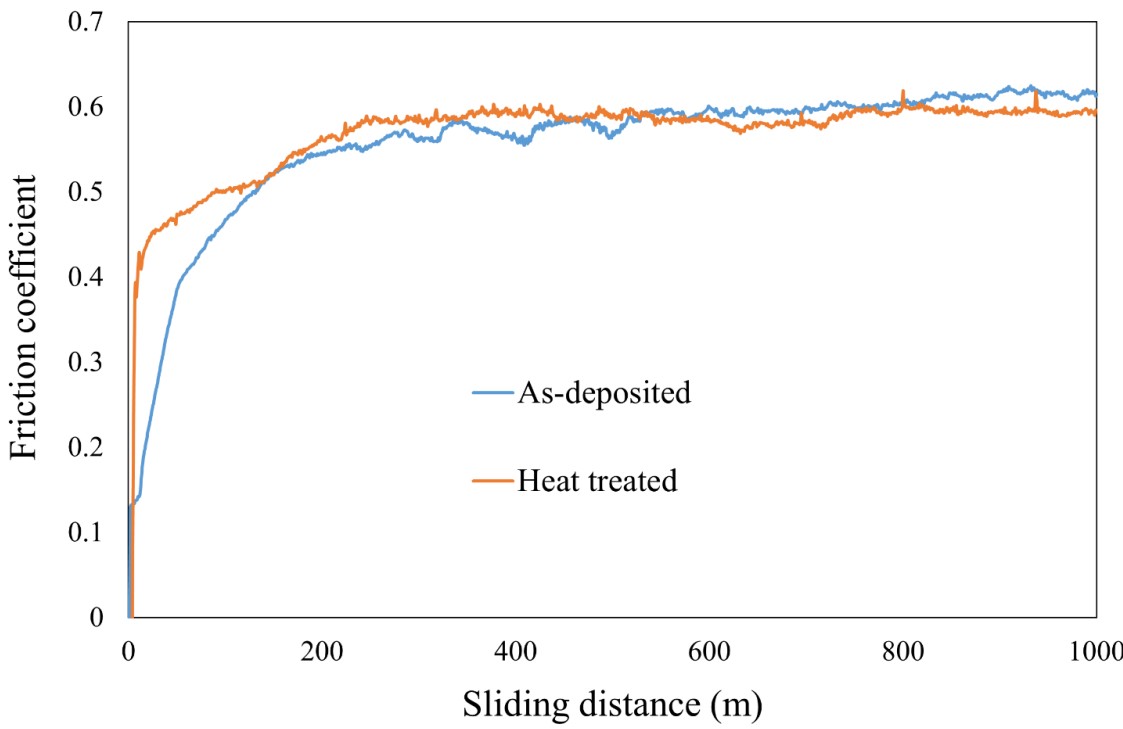

**Figure 10.** Evolution of friction coefficient of the as-deposited and heat-treated samples recorded during 1000 m of ball-on-flat sliding test.

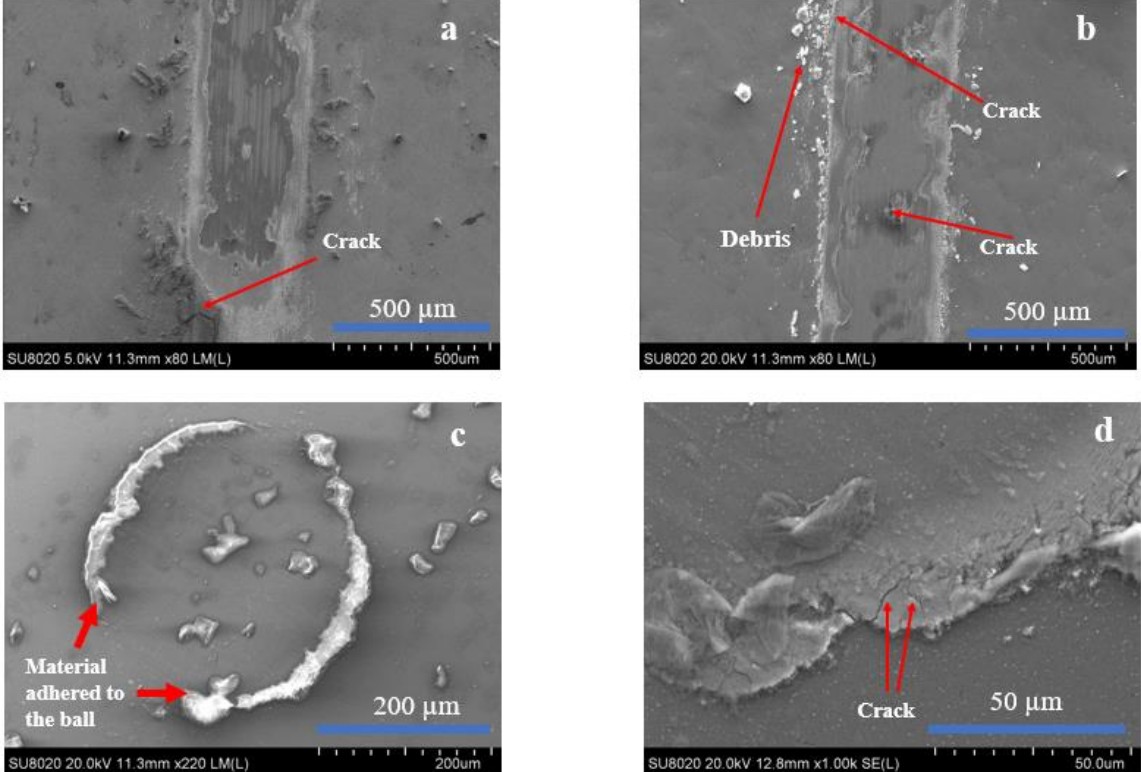

**Figure 11.** (**a**,**b**) SEM micrographs of the wear track of the heat-treated electroless nickel boron; (**c**,**d**) surface of alumina balls after the dry ball-on-flat sliding test.

**Table 7.** Chemical composition of the wear track on the heat-treated deposit, debris, and the ball after the dry ball-on-flat sliding test.

| Element | Oxygen (wt. %) | Aluminum (wt. %) | Iron (wt. %) | Nickel (wt. %) |
|---|---|---|---|---|
| Wear track on deposit | 4.9 | 0.5 | 0.6 | 92.2 |
| Debris | 33.8 | 4.5 | 0.1 | 61.06 |
| Ball | 51.1 | 21.8 | 0 | 24.1 |

In this wear test, as in the previous one, three different wear mechanisms were observed. Nevertheless, it is important to point out that the proportion of the adhesive mechanism to the abrasive one during the wear process, as well as the evolution of the friction coefficient value, will be strongly influenced by the amount of debris present in the wear track. At a sliding speed as low as 0.1 cm/s, the amount of debris removed by centrifugal forces during the continuous sliding system is small and the majority of debris will lie in the contact between the ball and the sample surface. In these conditions, due to the oxidation process that takes place, the proportion of the adhesive wear mechanism as compared to the abrasive one is higher. This phenomenon is corroborated by the evolution of the friction coefficient (see Figures 10 and 11) and the higher amount of oxygen found in both wear tracks (sample and ball) as compared to those encountered in the track corresponding to the ball-on-disk sliding motion (see Tables 6 and 7).

On the other hand, during reciprocating sliding, the amount of loose debris is kept in contact for a much longer time, and additionally, it is much higher for the reasons explained previously. In these conditions, it is obvious that a high proportion of an abrasive wear mechanism takes place, as compared to the adhesive one (see Figure 11a,b).

*3.8. Corrosion Properties*

3.8.1. Potentiodynamic Polarization Tests

The potentiodynamic polarization curves for the as-deposited and heat-treated coatings are presented in Figure 12. First of all, the corrosion potential value of the heat-treated deposit increased, and the shape of the curve was not changed after heat treatment, as compared to the as-deposited one. However, the corrosion current density of the heat-treated coating was higher than that of the as-deposited coating.

The most important parameters that impact the corrosion resistance of electroless nickel-boron coatings are chemical composition and homogeneity, the stress state in deposits, the presence of impurities, the structure and morphology of deposits, the presence of pores and pits, and the nature of the substrate and test medium [44,50].

The increase in corrosion density after heat treatment is attributed to the crystallization and formation and new phases. A change from an amorphous structure to a crystalline increase the grain boundary area per unit volume, which act as sites for corrosion attack. Different phases formed after heat treatment have different reduction potentials, which can generate a micro-galvanic cell [25,38,51,52]. Additionally, dislocations, which are detrimental to corrosion properties, are also formed after heat treatment [52]. The increase in the corrosion potential value is consistent with previous investigations [23,52,53]. This increase could be attributed to the insufficient stress release in the coatings during treatment at 300 °C.

3.8.2. Salt Spray Test

Neutral salt spray tests were carried out to characterize the corrosion properties of the heat-treated coatings. The salt spray test results are shown in Figure 13. The heat-treated coating started to corrode in a shorter time (after four hours); however, the as-deposited coating [35] showed corrosion marks after two days of immersion. After 10 days, 23% of the heat-treated coating surface was corroded, which is higher than that of the as-deposited coating (only 7.2% coating surface was corroded) [35]. These results are coherent with the

findings from the potentiodynamic polarization tests, indicating a clear decrease in the corrosion resistance of the coating. In addition, it needs to be mentioned that there is no evolution in the corrosion damage between four hours and seven days. This is due to the salt located in the gap between the coating and the seal.

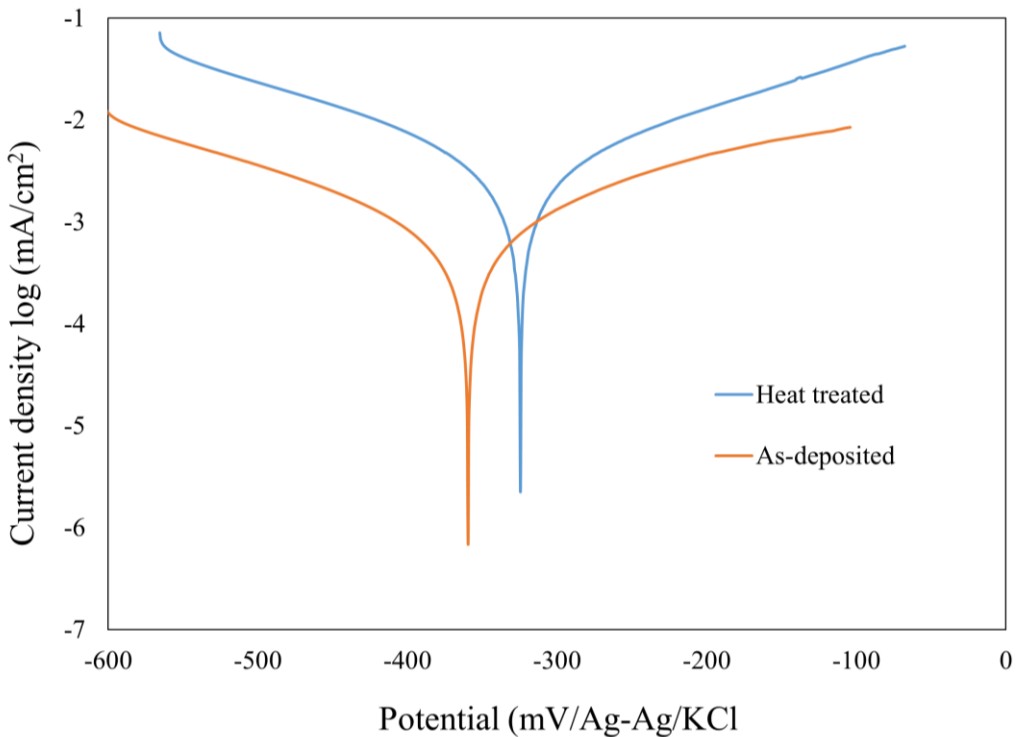

**Figure 12.** Potentiodynamic polarization curves of the heat-treated and as-deposited electroless nickel-boron deposit electroless.

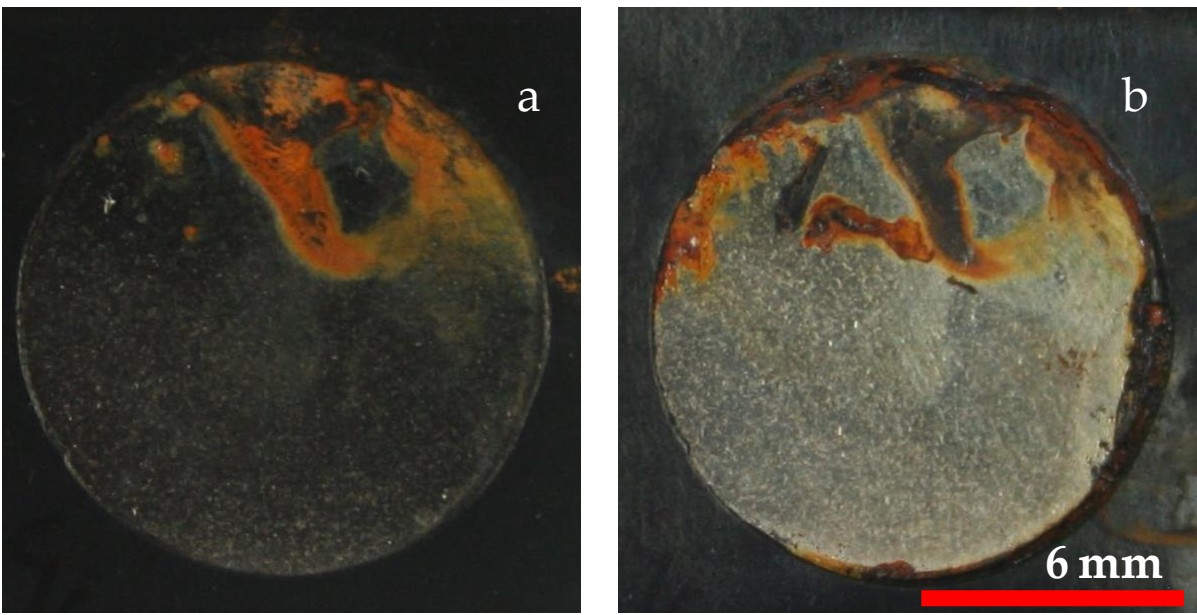

**Figure 13.** Heat-treated deposit images after 4 h (**a**) and 10 days (**b**) of salt spray test.

## 4. Conclusions

- The DSC trace of the as-deposited samples exhibits only one exothermic peak at around 284 °C, which is attributed to the formation of $Ni_3B$ phases, and heat treatment above 250 °C led to the formation of the $Ni_3B$ crystalline phase;
- The maximum hardness was obtained for heat treatment of 4 h at 300 °C and was 40% higher than the hardness of the as-deposited coating. This was confirmed by 3 different hardness measurement methods, and the results are $1196 \pm 120$ $hk_{50}$, $1277 \pm 181$ $hv_{50}$, and $16.2 \pm 3$ GPa;
- The morphological features of the nickel-boron coatings (limited roughness and featureless morphology) are unchanged after the heat treatment;
- Similar wear mechanisms (abrasive, adhesive, and fatigue), typical for the contact between a soft coating and a hard ball, were observed after both pin-on-disk and ball-on-flat sliding wear tests on the heat-treated coatings;
- The COF of the heat-treated coating during pin-on-disk sliding increases; however, it is the same during the ball-on-flat test in comparison with the one of the as-deposited coating. This difference results from the different proportions between adhesive and abrasive wear obtained after the tests;
- The wear track evaluation by SEM indicated that the coating maintained its integrity after both pin-on-disk and ball-on-flat sliding tests;
- The heat-treated coating exhibits a lower corrosion resistance after heat treatment, which is noticed from both potentiodynamic and salt spray tests.

**Author Contributions:** Conceptualization, V.V. and M.Y.; methodology M.Y.; software H.A.K.; validation, M.Y., A.M. (Alex Montagne) and V.V.; formal analysis, M.Y.; investigation M.H.S.; resources M.Y.; data curation, M.Y.; writing—original draft preparation M.Y. and V.V.; writing—review and editing A.M. (Alex Montagne) and M.H.S.; visualization A.M. (Alexandre Mégret); supervision A.M. (Alex Montagne) and V.V.; project administration V.V.; funding acquisition A.M. (Alex Montagne). All authors have read and agreed to the published version of the manuscript.

**Funding:** This study was supported by the INTERREG VA program and European Regional Development Fund (FEDER) in the framework of the AltCtrlTrans project.

**Institutional Review Board Statement:** Not applicable.

**Informed Consent Statement:** Not applicable.

**Data Availability Statement:** This study did not report any data.

**Acknowledgments:** The authors gratefully acknowledge INTERREG for funding. The authors also would like to thank Yoann Paint from Materia Nova for his help with the analysis of the samples by SEM. The author wants to show his appreciation to Sébastian Colmant-Midol for his great help with sample polishing and hardness measurements.

**Conflicts of Interest:** The authors declare no conflict of interest.

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
