# Peer review of "Selection of New Heat Treatment Conditions for Novel Electroless Nickel-Boron Deposits and Characterization of Heat-Treated Coatings"

_coatings, doi:10.3390/coatings13010001_

Round 1

Reviewer 1 Report

In the paper, the properties of electroless nickel-boron coatings were investigated. Although the manuscript is an interesting subject, some necessary improvements must be made in order to incorporate this work for publication.

1.    The quality of all images in the full text should be improved.

2.    Table 3 should be corresponding to the content described in the text.

3.    Fig. 9c and 9d show the worn surface morphology of aluminum oxide ball, which cannot indicate that the wear mechanism of the heat-treated deposit is fatigue wear. At the same time, the worn surface morphology of the as-deposited coating should be added to form a comparison.

Author Response

Dear reviewer;

First of all, I want to thank you for your review

  1. The quality of the all images have been improved.
  2. There was a typo, it was corrected. The change can be seen via track changes. 
  3. The coating undergoes wear fatigue as a consequence of the cyclic loading applied by the indenter during the test. The presence of cracks at the wear track (not only on material adhered on the alumina ball, as shown in Figures 9 (a, b) and 11 (a, b)) is the evidence of the occurrence of such a damage mechanism in the coated system. At the same time, the material adhered onto the alumina ball surface is also bound to undergo fatigue due to cyclic loading.

Reviewer 2 Report

This manuscript presentsSelection of new heat treatment conditions for novel electroless nickel-boron deposits and characterization of heat treated coatings, but minor changes have to be done before publication.

1.     Pictures in Figure 2 are so small and blurry, which should be replaced by some clear images.

2.     Figure 4,5 should be re-edited: captions are covered by two white color blockings.

3.     On Page 13: “Iron was detected on the wear track of the deposit but not on the ball indicating that the coating maintained its mechanical integrity.” The reason should be explained more clearly.

4.     In Figures 11, 12, and 13, the images are not professional, which should be re-corrected.

5.     Conclusions should be corrected to express the brief and core findings.

Author Response

Dear reviewer;

First of all, I want to thank you for your review.

  1. The quality of the Figure 2 was improved.
  2. Figure 4 and Figure 5 were re-edited and their quality was improved.
  3. Iron was detected on the wear track of the deposit but not on the ball, as a consequence of the low coating thickness. Therefore, the detected iron belongs to the substrate and the fact that it was not found on the ball indicates that the coating maintained its mechanical integrity during the wear test. 
  4. Figures 11, 12, and 13 were modified and their quality was improved.
  5. The conclusion part was modified to present brief and core findings.

Reviewer 3 Report

This manuscript investigate a modifying the concentration

of complexing and reducing agents of electroless NiB sulution. And the deposits obtained from this novel plating bath do not present the same properties as the standard electroless nickel-boron coatings:. This work has some new progress in this field, but the followting items must be improved to meet the publication requirements.

(1) There are some expressions which are needed to be revised, which I only mark the abstract part in the attachments.

(2) The shortages of the Pb stablizer and the performances inferior of form research is not clearly descripted in the introduction part.

(3) The quality of the figures are poor, the foamat is also needed to be uniformed.

(4) The performance in NSS test is not ideal for both kinds of coatings. 

(5) A comprehensive comparison with the traditional electroless NiB coating is needed to valid the proposed conclusions.

a

Author Response

Dear reviewer;

First of all, I want to thank you for your review. 

(1) The manuscript was modified based on your comments.

(2) The removal of lead stabilizer from electroless Ni-B baths has been amply described in previous research of our group that is cited in the introduction. As for the performances, the stabilizer -free have similar performances to all but the best nickel-boron coatings available at the present time so we did not feel the need to discuss this. 

(3) The quality of the all figures were modified. 

(4) Electroless nickel-boron coatings are the hardest and most wear resistant among all electroless nickel coatings. Their corrosion resistance is reputed to be lesser than electroless nickel-phosphorus coatings. However, the corrosion resistance of electroless nickel-boron coatings can be improved. Recently, we analyzed the coating produced in replenished bath and it presents a better corrosion resistance than the as-deposited and heat-treated electroless nickel-boron coatings. Those results will be published in a coming paper. We’d also like to point that, while NSS is considered as the industry standard for corrosion resistance, this test is not well suited for the evaluation of corrosion performance of metallic coatings.

5. In the previous study [35], the novel coating was comprehensively compared with the traditional electroless nickel-boron coating. The novel coating presents modified properties as compared to the electroless nickel-boron coatings: its surface is smooth and featureless, its boron content is lower and its corrosion resistance is improved while maintaining the same hardness. We voluntarily omitted the comparison of our as plated coating with the traditional coating from this paper to avoid redundancy with our previous paper. 

Reviewer 4 Report

Excellent written paper ! Very minor changes are recommended.

Author Response

Dear, 

The manuscript has been modified based on changes. 

Round 2

Reviewer 1 Report

Agree to accept the manuscript

Reviewer 3 Report

The manuscript has been well revised, the reviewer has no more comment.